# Endoscopic Endonasal Approach in Craniopharyngiomas: Representative Cases and Technical Nuances for the Young Neurosurgeon

**DOI:** 10.3390/brainsci13050735

**Published:** 2023-04-28

**Authors:** Jorge F. Aragón-Arreola, Ricardo Marian-Magaña, Rodolfo Villalobos-Diaz, Germán López-Valencia, Tania M. Jimenez-Molina, J. Tomás Moncada-Habib, Marcos V. Sangrador-Deitos, Juan L. Gómez-Amador

**Affiliations:** Department of Neurosurgery at National Institute of Neurology and Neurosurgery “Manuel Velasco Suárez”, Mexico City P.C. 14260, Mexico; jaragon@innn.edu.mx (J.F.A.-A.); ricardomarian@neurocirugia-innn.com (R.M.-M.); rvillalobos@innn.edu.mx (R.V.-D.); glopezv@innn.edu.mx (G.L.-V.); tjimenezmolina@innn.edu.mx (T.M.J.-M.); jmoncadahabib@innn.edu.mx (J.T.M.-H.); msangrador@innn.edu.mx (M.V.S.-D.)

**Keywords:** craniopharyngioma, endoscopic, infundibulum

## Abstract

Craniopharyngiomas (CPs) are Rathke’s cleft-derived benign tumors originating most commonly in the dorsum sellae and representing 2% of intracranial neoplasms. CPs represent one of the more complex intracranial tumors due to their invasive nature, encasing neurovascular structures of the sellar and parasellar regions, making its resection a major challenge for the neurosurgeon with important postoperative morbidity. Nowadays, an endoscopic endonasal approach (EEA) provides an “easier” way for CPs resection allowing a direct route to the tumor with direct visualization of the surrounding structures, diminishing inadvertent injuries, and providing a better outcome for the patient. In this article, we include a comprehensive description of the EEA technique and nuances in CPs resection, including three illustrated clinical cases.

## 1. Introduction

Craniopharyngiomas (CPs) are benign intracranial extra-axial tumors (OMS grade I) that originate from remnants of the Rathke’s cleft, representing 2% of all intracranial neoplasms, with an estimated incidence of 0.17 to 0.2 and prevalence of 4.78 per 100,000 [1].

Surgical management of CPs is challenging because of the vicinity to critical neurovascular structures, demanding a thorough understanding of the anatomy of the suprasellar region. The extension of the tumor in relation to the optic chiasm, pituitary gland and stalk, hypothalamus, carotid artery, and anterior cerebral artery complex is essential for surgical planning. Surgical options include transcranial and endonasal endoscopic approaches (EEA) [2].

EEA provides a direct route to the sellar region, with improved midline exposure without retraction of brain parenchyma and neurovascular structures, obtaining a better visualization. This approach is ideal for lesions without significant lateral growth and retrosellar CPs with suprasellar third ventricular extension [3,4].

The aim of this study is to familiarize young neurosurgeons with the anatomy of the sellar and suprasellar region, as well as the advantages of EEA in the resection of craniopharyngiomas. This remains paramount as endoscopic resection techniques have become an accessible option for all neurosurgeons, making it necessary to master dissection techniques to preserve critical neurovascular structures when facing CPs.

## 2. Materials and Methods

A comprehensive description of the surgical EEA in CPs is deeply analyzed by summarizing the technique and detailing a step-by-step approach based on the senior author’s experience in our hospital. For its comprehensive description, the technique was divided into nasal, sphenoidal, sellar, and closure phases. Three illustrative clinical cases are included.

## 3. Results

### 3.1. Endoscopic Endonasal Approach

In the operative room, the patient is positioned supine with the head tilted 15 degrees away from the surgeon so the nasal fossae are facing the endoscope’s trajectory. The trunk is elevated 20 to 30 degrees in order to aid venous return. The patient’s head is in neutral position when our target is the sellar region, and extended 10 to 30 degrees or flexed 20 to 40 degrees when the target is located in the anterior fossa or the clival region, respectively. Perioperative steroids (100 mg hydrocortisone) are administered, as well as a single dose of broad-spectrum antibiotics. Topical nasal decongestant is employed in order to reduce nasal bleeding. It is important to consider preparing the abdominal wall and outer thigh in an antiseptic manner since abdominal fat and fascial graft may be used for reconstruction [5,6].

The EEA is divided into four phases: nasal, sphenoidal, sellar/parasellar, and closure.

#### 3.1.1. Nasal Phase

A 0-degree endoscope is introduced into one nostril to identify the relevant anatomy (superior, middle, and inferior turbinates laterally, nasal septum medially, and the choana posteroinferiorly) [7]. The superior and middle turbinates are landmarks to identify the sphenoid ostium and both turbinates are coagulated and lateralized with blunt dissection, avoiding mucosal injury. The sphenoid ostium is identified 1.5 cm above the choana. A nasoseptal flap is harvested, as described elsewhere [8], if needed. The flap can be harvested at the beginning of the procedure, or after tumor resection in case of cerebrospinal fluid (CSF) leak, and should be tucked into the choana for protection during the operation [9]. The next step is to expose the sphenoidal rostrum, removing the mucosa in order to detach the nasal septum and vomer with a dissector. About 1.0 to 1.5 cm of the posterior nasal septum is removed for simultaneous access to the sphenoid sinus through both nasal nostrils.

#### 3.1.2. Sphenoidal Phase

The anterior wall of the sphenoid sinus is enlarged circumferentially, preserving the sphenopalatine artery which is located inferolaterally [2]. The sphenoid rostrum is removed using Kerrison rongeurs and high-speed drills. It is imperative to know the anatomy of the sphenoid sinus and its variants: “sellar”, “pre-sellar”, and “conchal” types (Figure). Pre-sellar and conchal variantes are not absolute contraindications to perform an EEA and, in these cases, the bone is removed through careful drilling and employing neuronavigation. Mucosa within the sphenoid sinus is removed to reduce the risk of postoperative mucocele [10]. There are septations inside the sphenoid sinus that must be removed carefully, being aware that 20% of septations lead to a cavernous carotid protuberance [7].

#### 3.1.3. Sellar Phase

A 4 mm diamond burr is preferred to make an initial opening of the pituitary fossa, as the sellar floor can be thin and partially dehiscent because of chronic remodeling by a large intrasellar mass. The durotomy is usually carried out with a sickle or retractable knife in a cruciate fashion, starting in the middle sector and extending it with angled microscissors in a cruciform fashion. After the dura is opened and hemostasis is achieved, exploration of the intrasellar mass depends on the nature of the pathologic process. For a large mass, as CPs, with a cystic and solid component, resection begins in the inferior and lateral portions of the tumor to allow the superior aspect to descend into the surgical field at last. If the superior portion is delivered first, diaphragmatic descent will obscure the operative field. If the tumor has a suprasellar component, the diaphragm can be sharply dissected and incised. Most CPs are soft in consistency and the resection is usually performed with a variety of microdissectors, ringed curettes, and suction cannulas. On the other hand, if the tumor has a harder consistency, it can be removed with the aid of an ultrasonic aspirator. After tumor removal, exploration of the surgical field with the 0-degree or 30-degree endoscope is mandatory to look for any residual tumor [11].

#### 3.1.4. Closure

In case of a CSF leak, there are two well-described techniques: the gasket seal and the multilayered technique. The gasket seal method consists in placing a piece of allograft dural substitute over the bony defect so that its dimensions exceed that of the defect by at least 1 cm circumferentially. A rigid implant cut to fit the opening is then placed over the dural substitute and counter-sunk within the bony defect [12,13]. The multilayered reconstruction consists of placing layers in apposition to one another, the first being an inlay dural substitute, followed by an onlay fascia lata graft, thereby potentially obviating the need for a rigid buttress [13]. The nasoseptal flap is placed over the preferred method so that the flap is in direct contact with the surrounding bony skull base and is subsequently held in place with fibrin glue. In cases with no evident CSF leak, a free mucosal graft from the middle turbinate or nasal floor can be placed alone over the surgical cavity and held in place with absorbable nasal packing [14].

### 3.2. Representative Clinical Cases

#### 3.2.1. Case 1

A 24-year-old man with a previous history of incomplete transcranial resection of a CP was admitted to the emergency department with a 3-month history of progressive visual loss, nausea, and vomiting. Neurological examination revealed a Glasgow Coma Scale of 14, pupils of 4 mm with poor light response, and no motor or peripheral sensory deficits. Endocrinological examination was relevant only for central hypothyroidism. An MRI was performed (Figure 1), revealing a T1 hypointense and T2 hyperintense large sellar and suprasellar cystic, lobulated lesion, with significant upward displacement of the third ventricular floor. An EEA was performed on this patient, using the technique described previously (Figure 2 and Appendix A). Following a dural opening, the cyst wall was punctured, releasing a motor-oil-like liquid content. After cyst drainage, meticulous debulking of the solid component of the tumor with ringed curettes was performed. Finally, the sellar floor was reconstructed in a multilayered fashion. Postoperative CT showed no evident residual tumor. Postoperatively, the patient developed transient diabetes insipidus (DI), which was satisfactorily managed with oral desmopressin, and he was discharged from the hospital on the fifth postoperative day.

#### 3.2.2. Case 2

A 34-year-old woman with complaints of headache and bilateral loss of visual acuity in the last year, presented to our emergency department due to acute onset of gait disturbance and sleepiness. Upon arrival, an urgent CT scan was performed, revealing an isodense mass in the sellar and suprasellar region with calcifications, conditioning obstructive hydrocephalus. An emergent ventriculoperitoneal shunt was placed. Endocrinological testing revealed low levels of thyroid-stimulating hormone, cortisol, and free thyroxine, so hormonal replacement therapy was initiated before surgical treatment was deemed safe. Brain MRI revealed a hypo and hyper-intense lesion on T1 and T2-weighted MRI, respectively, compatible with a cystic lesion located in the sellar region and extending upward into the third ventricle with brainstem displacement (Figure 3). The patient underwent resection of the lesion by an endoscopic extended transplanum–transtuberculum approach as shown in Figure 4 and Appendix A. The patient developed DI postoperatively and received subcutaneous desmopressin. No cerebrospinal fluid leakage was observed postoperatively. A vision assessment 6 months postoperatively showed no changes in visual acuity.

#### 3.2.3. Case 3

A 19-year-old man was referred to our hospital for endocrinological evaluation due to delayed pubertal development. Hormonal tests were performed revealing low testosterone, thyroxine, and cortisol levels. The patient reported a 2-month history of asthenia, polyuria, polydipsia, and blurred vision. Neurological examination was relevant for decreased bilateral visual acuity and bitemporal hemianopsia. Brain MRI revealed a sellar and suprasellar hypointense lesion on T1W, mixed iso and hyperintense lesion on T2W and FLAIR, and heterogeneous enhancement on postcontrast sequence, suggestive of a cystic lesion with a solid sellar component (Figure 5). The patient underwent transsphenoidal endoscopic resection of the lesion wranssellarsellar approach as shown in Figure 6 and Appendix A. After opening the cyst wall, the solid component of the lesion was drained, which had a greenish and muddy appearance, but was otherwise easily aspirable. The solid intratumoral content was removed using ringed curettes and suction, followed by capsule mobilization and sharp extracapsular dissection employing pituitary rongeurs and microscissors. Finally, a multilayered reconstruction of the sellar floor was performed. The patient had an uneventful postoperative course and, during the follow-up appointment, he reported significant vision improvement. Histopathologic evaluation reported an adamantinomatous CP and the patient was referred to radiosurgery for adjuvant treatment.

## 4. Discussion

CPs represent quite challenging lesions that require multidisciplinary management. What has been theorized and later proven to be the advantage of the extended endonasal endoscopic approach (EEEA) for resection of CPs is that gross total resection (GTR) is more achievable with this technique, having ranges of up to 70% in some series [3,15,16]. One of the key factors to determine whether an EEEA is feasible is preoperative evaluation of the lesion, with many details to pay attention to. However, many authors have consistently described the most important characteristics as the position of the optic chiasm (OC), the pituitary stalk (PS), and the invasion of the lateral compartments [2,17,18]. Many classifications have been suggested for surgical approach decisions, being one of the most iconic the one proposed by Kassam and the Pittsburgh group in 2008, dividing these lesions into four categories, according to the relation to pituitary stalk and naming grade IV the lesions that are exclusively in the third ventricle [17]. These grade IV lesions have been suggested to be better reached through transcranial approaches. Although some authors are exploring the capability of doing it purely endonasal in a safe manner, this remains one of the limitations of an EEEA for the treatment of purely intraventricular CPs [17]. Recent publications affirm that retrochiasmatic CPs with extension to the third ventricle can be successful reached via EEA; however, as we previously mentioned, this statement should be taken carefully. From this perspective, the European Association of Neurosurgical Societies still recommends the transcranial approach for intraventricular CPs [19,20,21].

One of the key considerations for surgeons in their learning curve is to divide the invasion of the tumor into infra and supra diaphragmatic compartments. Originally, only tumors in the infradiaphragmatic compartment were removed with the EEA but, as more experience was obtained, the supradiaphragmatic lesions started to be treated this way [22]. Doing a thorough analysis of the preoperative MRI is key to making the most optimal decision for the approach.

Understanding each stage during the endonasal approach is essential for the young neurosurgeon. The nasal stage is usually performed by an ENT surgeon in most centers, although the anatomical knowledge and technique mastering is a great ability to achieve as a neurosurgeon. CSF leak is one of the most prevalent complications of these approaches and, historically, the use of nasoseptal flap (NSF) had proven to reduce CSF leak rates in all kinds of endonasal approaches [8,23].

For lesions extending to the suprasellar compartment, we must try to harvest an NSF from the beginning of the nasal step. This NSF is usually harvested from the side which will undergo less manipulation depending on the lateral invasion of the tumor. As we mentioned, the NSF is a crucial step in the prevention of CSF leak and meningitis hence for its correct realization technique must be well understood and dominated by the surgeon [23,24].

During the sphenoidal phase, widening the opening on the sphenoid rostrum is an advisable behavior with the goal to attain freedom of movement in the field with the tools. This will strongly depend on the sinus pneumatization but should not be a limitation to accomplishing a sufficient opening and a comfortable setup. Anatomical knowledge plays a crucial role and landmark identification will consist the main goal of the sinus stage in which the visualization of the sella turcica, tuberculum sellae, planum sphenoidale, and other landmarks, gives the surgeon the confidence and safety needed to continue with the approach [25]. Some less pneumatized sinuses will offer a harder challenge for less experienced surgeons. In these cases, neuronavigation may come as a very useful tool.

In the sellar stage, opening the sella turcica is usually done with a high-speed drill to thin the bone and after making a small opening, the use of angled dissectors may constitute a safe way to continue with the already thinned bone removal. Approaching the lateral limits of the boney opening, rongeurs are usually used taking special care to remove these portions with small bites avoiding the use of excessive force, further drilling may be used to thin the bone and see through the structures behind these lateral portions of the bone removal. The visualization of the anterior compartment of the cavernous sinus should be the goal permitting early coagulation but this can be tailored for each specific tumor invasion [3].

The goal is to find a safe and direct route to the tumor. Usually, the tumor is not adherent in its whole circumference and classical CPs show most adherent parts in the hypothalamus, which is to be expected when taking into consideration the origin of the lesion. Even when preoperative hypothalamic disfunction exists by tumor compression, this part of the lesion should be always evaluated with direct sight of the adherences, and should be dissected when possible, either bluntly or preferably sharply, since sharp dissection needs the least amount of traction to bring the lesion into the field. Sometimes, it is preferable to leave a small tumor capsule cuff when none of these techniques can be performed in a safe manner. This point can be debatable as some authors prefer to reach GTR if any preoperative sign of hypothalamic involvement exists [26].

Retracting or manipulating the tumor is thought to be a risky maneuver but it is nonetheless unavoidable in some instances, when this situation is presented, it is advisable to never pull the tumor without pushing the tissue you are trying to separate it from. This creates a pivot point that is usually in the field and on sight, rather than a blind spot that can be an important structure (i.e., carotid or hypothalamus). We usually call this maneuver traction–countertraction.

It has been proposed in the literature a useful classification for endoscopic endonasal surgery according to the degree of technical complexity, taking into account two main factors: the affected compartment (for example, pituitary fossa or cavernous sinus) and the pathology in question, as shown in Table 1 [27,28]. As can be seen, although craniopharyngiomas are classified in grade II of technical complexity when they are limited to the pituitary fossa, their invasive nature could classify them in a higher grade when they extend to the interpeduncular cistern. This classification must be known and used by young neurosurgeons in the planning of resection of these complex tumors, permitting them to visualize the anatomical corridors and associated structures compromised by the tumor and prepare the necessary endoscopic equipment for the procedure.

Accordingly, young neurosurgeons must keep in mind that these approaches require a steep learning curve, which may be associated with an increase in complication rates as the surgeon obtains experience. Therefore, to minimize surgical morbidity, it is highly advised to acquire surgical experience and technical competency from less complex pathologies in a step-by-step fashion before proceeding to the most technically demanding ones [27,28]. As previously noted, the technical complexity and degree of expertise needed for endoscopic resection of craniopharyngiomas are highly variable depending on the tumor location, extension, and relationships with neurovascular structures [3,16,20]. Consequently, we have found it useful to divide the endoscopic endonasal approaches for craniopharyngiomas into different categories of progressive technical complexity and expertise required, as initially proposed by Baldauf et al. [3] and subsequently modified in the present article Table 2.

A suggested strategy is to first acquire experience with intrasellar and intra-suprasellar lesions before moving to more complex procedures such as suprasellar craniopharyngiomas (Category C), often requiring an extended approach, intradural tumor dissection and exposure of neurovascular structures. Additionally, the importance of laboratory training with anatomical models and cadaveric dissections cannot be over-emphasized, as well as assisting/observing experienced surgeons before attempting these procedures. Finally, it is essential for young neurosurgeons to recognize their limitations and to not hesitate in performing a transcranial approach when the case complexity is beyond their endoscopic capabilities [27,28].

## 5. Conclusions

CPs continue to be a high-complexity intracranial pathology despite advances in neurosurgery, nevertheless as EEAs evolve this technique offers greater advantages over the conventional transcranial approaches, allowing a better visualization of the tumor and its relationships with neurovascular structures surrounding it, thus contributing to achieve a GTR whenever possible. A comprehensive knowledge of this technique and its nuances, in conjunction with intensive laboratory training, is fundamental for young neurosurgeons who aim to perform these approaches. Finally, it should be noted that these approaches require a long learning curve, hence, the acquisition of surgical skills and experience from less complex cases is imperative before moving to the most challenging ones, and should be recognized that some cases may be better managed transcranially if the surgeon feels more comfortable and experienced with this approach.

## Figures and Tables

**Figure 1 brainsci-13-00735-f001:**
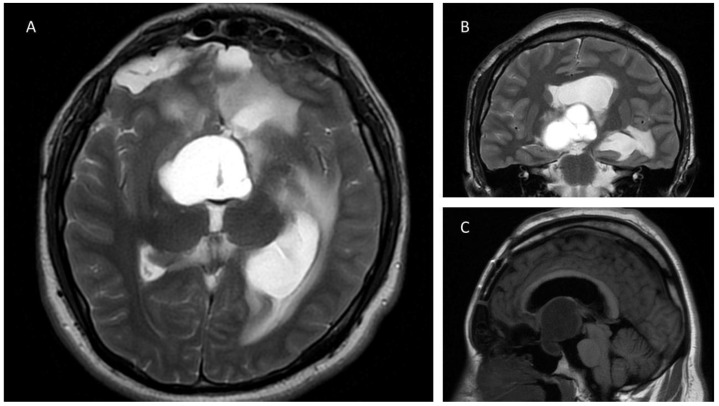
Axial (**A**) and coronal (**B**) T2-weighted MRI demonstrates a sellar lesion with suprasellar extension composed predominantly of a T2 hyperintense cystic component. Image (**A**) also shows dilatation of the left temporal horn of the lateral ventricle and postsurgical changes in both frontal lobules. In Image (**B**) the cystic and lobulated features of the lesion are seen. (**C**): Sagittal T1 weighted MRI image shows a hypointense cystic sellar lesion with upward displacement of the third ventricle.

**Figure 2 brainsci-13-00735-f002:**
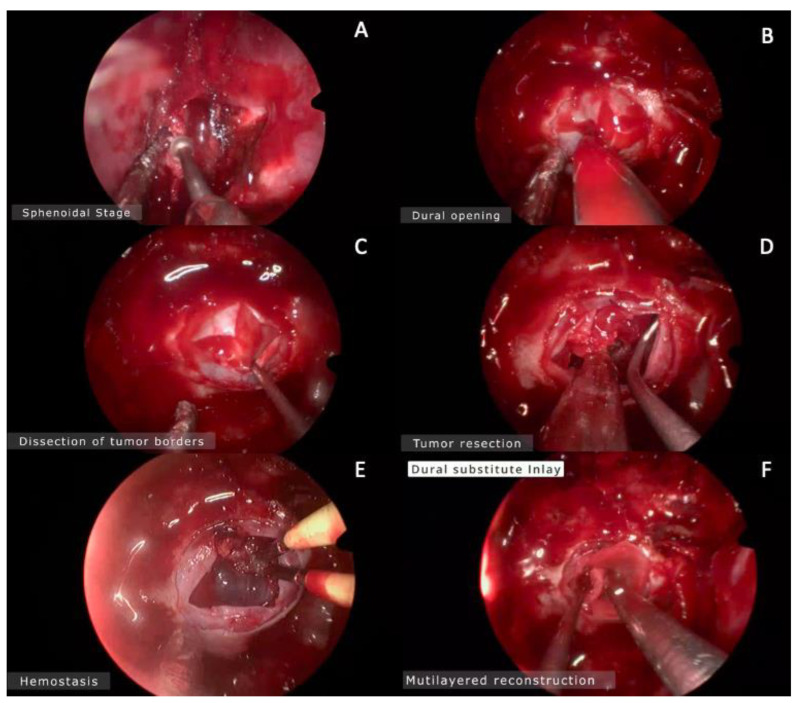
Intraoperative images. (**A**) Drilling the sphenoidal rostrum. (**B**) Dural opening in cruciform fashion using no. 11 blade. (**C**) Dissection of the tumor borders away from the dura mater using a fine microdissector. (**D**) Tumor resection begins from the inferior and lateral components in order to avoid the superior component of the tumor that obstructs the surgeon’s view. (**E**) Hemostasis. (**F**) Multilayered reconstruction.

**Figure 3 brainsci-13-00735-f003:**
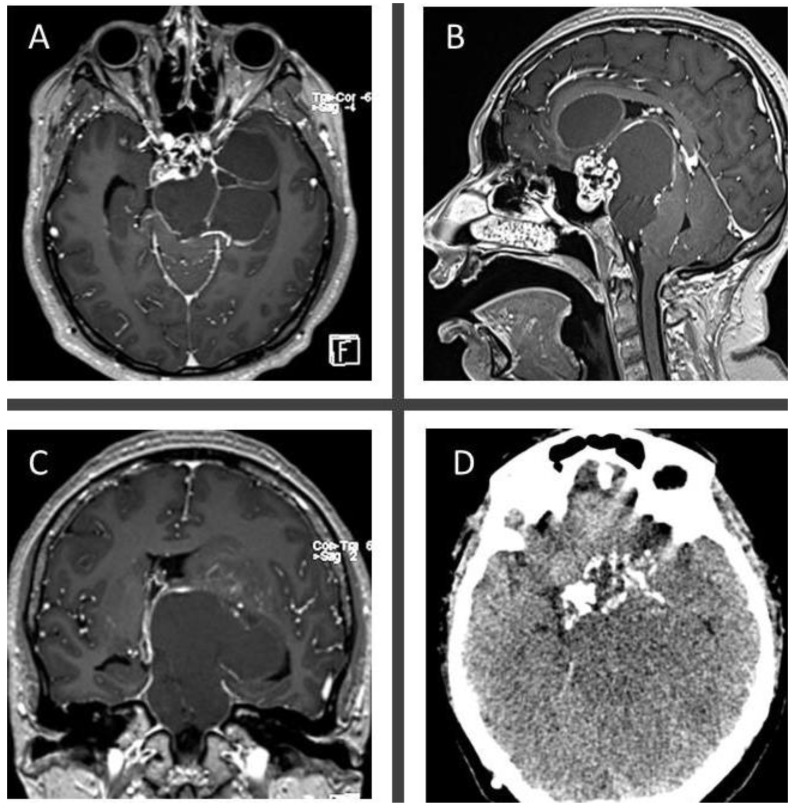
Axial (**A**), sagittal (**B**), and coronal (**C**) T1-post contrast MRI images show a mixed cystic-solid mass with a solid sellar, heterogeneously enhancing component, associated with a massive suprasellar and parasellar cystic, peripherally enhancing component extending upward into the third ventricle and displacing backwardly the brainstem. Axial, non-contrast CT (**D**) shows a hypodense sellar mass associated with massive peripheral calcifications.

**Figure 4 brainsci-13-00735-f004:**
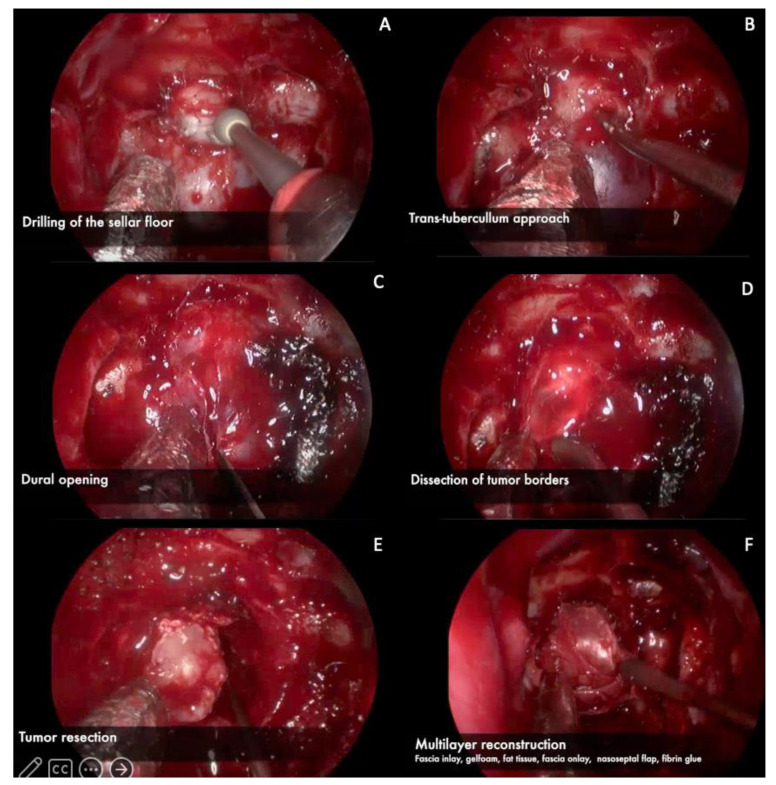
Intraoperative images. (**A**) Drilling of the sellar floor using a diamond drill. (**B**) In this case it was necessary to remove the tuberculum sellae because of the tumor size, using a diamond drill and rongeurs. (**C**) Dural opening in a cruciform manner. (**D**) Dissection of tumor borders away from the durI (**E**) Tumor resection. (**F**) Multilayer reconstruction (In-lay fascia, gelfoam, fat tissue, On-lay fascia, nasoseptal flap, and fibrin glue).

**Figure 5 brainsci-13-00735-f005:**
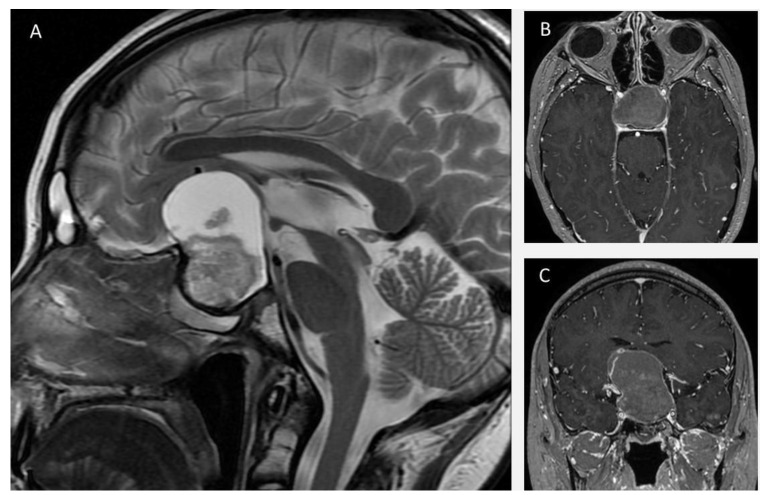
Sagittal T2-weighted MRI image (**A**) shows a large sellar and suprasellar mass with components of different signal characteristics. An isointense component relative to brain parenchyma is located predominantly in the sellar region, which is associated with an hyperintense cystic component in its superior aspect. Axial (**B**) and coronal (**C**) T1-post contrast MRI image reveals a sellar and suprasellar mass, with areas of peripheral and central enhancement. In image (**C**), a constriction of the mass at the level of the diaphragma sellae is seen.

**Figure 6 brainsci-13-00735-f006:**
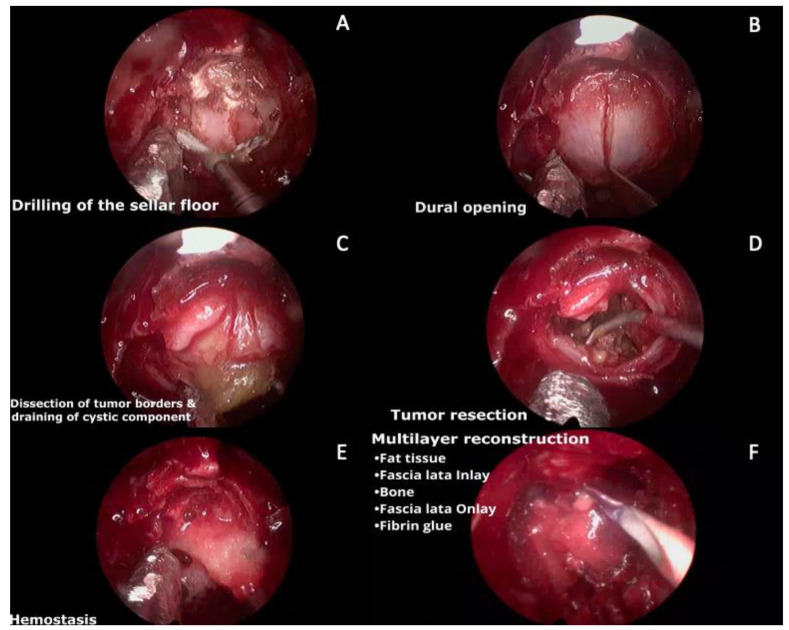
Intraoperative images. (**A**) Drilling of the sellar floor with a diamond drill. (**B**) Dural opening in a cruciform manner using a no. 11 blade. (**C**). Dissection of the tumor borders away from the dura, and, in this case, the cystic component was opened and suctioned. (**D**) Tumor resection of the lateral and superior parts until identification of the arachnoid layer Ip. (**E**) Hemostasis. (**F**) Multilayer reconstruction (fat tissue inside the sella, in-lay fascia lata, bone (gasket technique), on-lay fascia lata, and fibrin glue).

**Table 1 brainsci-13-00735-t001:** CSF. Cerebrospinal fluid, ACo. Anterior communicating artery.

Modified Level of Complexity in Endoscopic Endonasal Surgery		
Level	Compartment	Pathology
I	Extradural	CSF leak Chordoma Carcinoma
II	Pituitary fossa	Pituitary adenoma Craniopharyngioma
III	Anterior skull base floor	Meningioma Esthesioneuroblastoma Fibrous displasia
IV	Orbit Pterygo-pallatine fossa Maxillary sinus	Trigeminal Schwannoma
V	Cisterns Interpeduncular fossa Clivus	Craniopharyngioma Chordoma Pituitary adenoma
VI	Cavernous sinus	Pituitary adenoma Hemangioblastoma Meningioma
VII	Vascular	Paraclinoid aneurysm ACo aneurysm Basilar aneurysm Vertebral aneurysm
VIII	Intrinsic	Cavernoma Metastasis Glioma

**Table 2 brainsci-13-00735-t002:** Modified from Baldauf et al. [3]. References [16,17,20,22,27,28,29,30].

Categories of Technical Complexity and Surgical Expertise of Endoscopic Endonasal Approaches for Craniopharyngiomas		
**Category**	Tumor Location/ Extension	Technical Nuances
**A**	Intrasellar/infradiaphragmatic	Anatomic relationships similar to that of pituitary adenomas. The sellar floor is enlarged providing enough space for resection and, thus, facilitating the approach.
**B**	Intra-suprasellar/infradiaphragmatic	Neurovascular structures in the suprasellar side of the tumor protected by the diaphragm. Sufficient space between pituitary gland and the optic chiasm for tumor resection.
**C**	Suprasellar/preinfundibular or transinfundibular	An extended endoscopic endonasal approach is required. Size of the sella is often normal or reduced, resulting in a narrower approach due to closeness of the two intracavernous carotids. Arteries of the circle of Willis are often displaced by the suprasellar mass.
**D**	Suprasellar/retroinfundibular	Requires a more posterior approach. Stretch relationship with important neurovascular structures such as the mammillary bodies and basilar apex. Requires a pituitary transposition, increasing the technical complexity of the approach.
**E**	Arising or extending into the third ventricle	Risk of injury to the hypothalamus, brainstem, and other important neurovascular structures. Increased risk for CSF fistula due to communication of the ventricular cavity with the sphenoid sinus.

## Data Availability

Not applicable.

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
