# Peer review of "Endoscopic Endonasal Approach in Craniopharyngiomas: Representative Cases and Technical Nuances for the Young Neurosurgeon"

_brainsci, 2023, doi:10.3390/brainsci13050735_

Round 1

Reviewer 1 Report

The present study about the EEEA to manage craniopharyngiomas is well written and documented; the authors also added few very nice and didactic videos and they should be commended for their surgery and results; on the other hand the study does not add neither new knowledge nor new management strategies to the existing body of literature thus, in my opinion, the main message of this study should be better reoriented; I believe that the study could be greatly improved if the authors would add: 1) a table showing all the equipment (from basic to more sophisticated) to be available in “an endoscopic suite”; 2) a table showing all the advantages and disadvantages of the EEEA and the transcranial route; 3) a clinical algorithm to guide practitioners and especially young surgeons in their routine practice to choose the best way in managing such challenging pathology; 4) the case description is too long please shorten it; 5) as the authors state in the methods section the technique reported and discussed is based on the experience of the department’s senior author; the latter constitute a key feature in the way that, I believe it is mandatory for the study message to explain the step by step progression of this surgery through the different procedure in terms of number and difficulties to go through the learning curve before to approach by EEEA such pathology (please add a small paragraph in discussion or in the methods section).   

Author Response

Response to reviewer

The study messagge has been reoriented, emphasizing the long learning curve associated to these approaches and a step by step learning protocol has been suggested. Additionally two tables (Table 1 and 2) have been added to point out the different levels of complexity of the endoscopic endonasal surgery procedures. The extention of the clinical cases has been reduced to be more concise.

Reviewer 2 Report

Well written interesting overview concerning the endoscopic endonasal approach for CP. Alle aspects all well described en documented.

Case 3 is the most convincing indication for safe endonasal resection.

Case 2 is much more challenging concerning extension retroclical and left lateral; In the discussion lacks the potential risks/contra-indication of suprasellar lesions that a. cross nerves or b. cross vessels. This aspect should be mentioned to young colleaques starting with this technique. The video shows that a partial resection was achieved on Ct. That confirms the contra-indications a. en b.  as mentioned.

Case 1 shows resection of cystic lesion.

You should delineate a little more on the risks of suprasellair lesions crossing nerves or vessels in order to warn young colleaques not to pull tumortissue beyond these anatomic lines. Especially not in CP.

Also the endonasal resection of recurrent CP might be much more dangerous because of adhesions to third ventricle structures.

Author Response

Response to reviewer

Point 1: a table (Table 2) describing the technical nuances associated to the tumor location and extension has been added to guide the potential risk associated with the characteristics of the tumor.

Reviewer 3 Report

The Authors in this paper reported a case cases and technical nuances for the young neurosurgeon. I appreciated the effort of the authors in the detailed drafting of the article

After careful review, I would suggest some items to check:

- I appreciate the supplementary files with operative video, but I suggest inserting pictures inside the manuscript to improve young neurosurgeon's learning role.

- I decision-making about the better approach should be explained. The endoscopic endonasal approach is one of the ways to approach the CPs. I suggest inserting a decision-making algorithm.

- English language needs a revision.

page1image66696816  

Author Response

Response to reviewer

Point 1: pictures of the surgical procedure has been added inside the manuscript.

Point 2: the decision making rationale has been emphasized

Point 3: english lenguage has been further revised.

Thanks!

Round 2

Reviewer 1 Report

The authors answered almost satisfactorily to the reviewers comments. It is still missing a clear and schematic clinical/surgical algorithm otherwise now the study has been pretty improved.

Wg